# Level of Secretion and the Role of the Nerve Growth Factor in Patients with Keratoconus before and after Collagen Fibre Cross-Linking Surgery

**DOI:** 10.3390/ijms25010366

**Published:** 2023-12-27

**Authors:** Magdalena Krok, Ewa Wróblewska-Czajka, Olga Łach-Wojnarowicz, Joanna Bronikowska, Zenon P. Czuba, Edward Wylęgała, Dariusz Dobrowolski

**Affiliations:** 1Chair and Clinical Department of Ophthalmology, Faculty of Medical Sciences in Zabrze, Medical University of Silesia, Panewnicka 65 Street, 40-760 Katowice, Poland; ewaw8@wp.pl (E.W.-C.); olga.lachwojnarowicz@gmail.com (O.Ł.-W.); wylegala@gmail.com (E.W.); dardobmd@wp.pl (D.D.); 2Ophthalmology of Department, District Railway Hospital, 65 Panewnicka Street, 40-760 Katowice, Poland; 3Ophthalmology of Department with Paediatric Unit, St. Barbara Hospital, 41-200 Sosnowiec, Poland; 4Department of Microbiology and Immunology, Faculty of Medical Sciences in Zabrze, Medical University of Silesia, 40-055 Katowice, Poland; jbronikowska@sum.edu.pl (J.B.); zczuba@sum.edu.pl (Z.P.C.)

**Keywords:** β-NGF, keratoconus, cross-linking

## Abstract

Background: The aim of this study was to analyse the concentration of the nerve growth factor (NGF-β) in patients with keratoconus (KC) who are undergoing collagen fibre cross-linking (CXL) surgery in order to better understand the pathogenesis of this disease and observe the molecular changes occurring after the procedure. Among many cytokines, β-NGF seems to play an important role in the healing processes of corneal damage. Therefore, its role in the regenerative process after CXL treatment may affect the course of treatment and its final results. Tear samples from 52 patients were collected in this prospective study. Additionally, the patients also had a number of tests performed, including corneal topography using optical coherence tomography. Flat (K 1), steep (K 2), cylindrical (CYL), and central corneal thickness (CCT) keratometry were assessed. The tear samples were collected, and other tests were performed before the CXL procedure and afterwards, during the 12-month follow-up period. The NGF concentration was measured using the Bio-Plex Magnetic Luminex Assay. Lower levels of NGF-β were detected in the KC patients than in the control group (*p* < 0.001). The day after the procedure, the NGF-β level was significantly lower (on average by 2.3 pg/mL) (*p* = 0.037) than before the procedure, after which, the level of the reagent increases, but only in the group with the advanced cone, one month after CXL it was significantly higher (*p* = 0.047). Regarding the correlation of NGF with topographic measurements, the following were found: NGF-β correlates significantly (*p* < 0.05) and positively (r > 0) with K1 before the CXL procedure; NGF-β correlates significantly (*p* < 0.05) and positively (r > 0) with K1 one month after CXL; NGF-β correlates significantly (*p* < 0.05) and positively (r > 0) with CYL nine months after CXL; and, after twelve months, NGF-β correlates significantly (*p* < 0.05) and positively (r > 0) with K2 and K1. Corneal sensitivity did not statistically and significantly correlate with the level of NGF-β secretion. Our study suggests that NGF may be crucial in the development and progression of KC as well as in the repair mechanisms after CXL surgery. Further research is needed on the role of NGF and other inflammatory biomarkers for rapid diagnosis and selection of targeted therapy in patients with keratoconus.

## 1. Introduction

Keratoconus (KC) is a common condition that leads to the thinning of the central and near-central parts of the cornea, causing it to take on a conical shape. This corneal ectasia results in a high level of astigmatism and, consequently, a significant deterioration in vision [1,2]. The pathogenesis of the disease is not yet fully understood, and it may be affected by genetic, environmental, biochemical, and risk factors such as allergies, eye rubbing, asthma, and eczema [3,4]. Despite the fact that keratoconus is classified as a non-inflammatory disease [2,5], many authors suggest otherwise [6,7,8,9,10,11,12,13,14,15,16]. It cannot be definitively stated that this hypothesis is true, but the imbalance in the secretion of inflammatory factors (cytokines, chemokines, and growth factors) suggests that it may be significant.

Studies of tears and corneas in patients with keratoconus have revealed altered levels of inflammatory markers, including the nerve growth factor (NGF). The NGF and its receptors play a significant role in the corneal and conjunctival wound-healing process and in the regeneration of corneal nerves by stimulating fibroblasts and keratocytes, the expressing and secreting of extracellular matrix components, angiogenesis, and myofibroblast differentiation [17,18,19].

According to A. Lambiase, the molecular basis of keratoconus is the complete absence of NGF TrkA receptor (TrkA NGFR) expression and decreased NGF and p75 NTR expression, which consequently cause the dysregulation of the NGF secretion pathway, resulting in reduced NGF secretion [20]. Other studies report that NGF influences disease progression both as an independent factor and in correlation with other factors [21,22].

In our study, we measured the level of nerve growth factor in patients with keratoconus and examined how the level of NGF-β changed after CXL surgery. Currently, there is only one study describing the level of NGF secretion after a cross-linking procedure [23]. Studies conducted after similar minimally invasive refractive procedures, including LASIK, confirm that regenerative mediators, including NGF, may play a crucial role in the eye surface repair process after injury [24,25]. The nerve growth factor human recombinant produced in *E. coli* is a non-covalently disulfide-linked homodimer non-glycosylated polypeptide chain containing two identical one-hundred-and-twenty-one amino acids with a molecular weight of two 13.6 kDa polypeptide monomers.

The Food and Drug Administration (FDA) has approved eye drops containing NGF, which hold promise in the treatment of neurotrophic keratitis and may prove to be an effective treatment in other ophthalmic conditions requiring corneal regeneration, such as keratoconus [26,27].

The aim of this study is to assess the profile of NGF-β secretion after CXL treatment and attempt to correlate these fluctuations with topographic and morphological parameters of the cornea, especially with changes in the sensory innervation of the cornea.

## 2. Results

A total of 52 individuals with progressive keratoconus and 20 healthy participants in the control group were included in the study. The observation study was carried out over a period of 12 months after CXL procedure.

### 2.1. NGF-β Concentration

The comparison of NGF levels in the study group before the procedure to the control group indicated a significantly higher NGF-β level in the control group (*p* < 0.001) than in the study group, and this was always the case regardless of age, gender, or disease severity (Table 1).

The analysis of NGF level changes over time in the study group after CXL is presented in Table 2 and in Figure 1. One day after the procedure, the β-NGF level was significantly lower (an average of 2.3 pg/mL; *p* = 0.037) than before the procedure, and then the results increased and stabilised around 3 months, but these values were not statistically significant.

When the group was divided by the degree of keratoconus severity, the NGF-β level one month after CXL was significantly higher (*p* = 0.047) in the severe cases (11.80 pg/mL) compared to the mild cases (5.64 pg/mL) and the moderate cases (6.58 pg/mL) (Figure 2).

### 2.2. Correlations of NGF-β with Corneal Topographic Parameters

NGF-β correlates significantly (*p* < 0.05) and positively (r > 0) with Kf before CXL procedure; β-NGF correlates significantly (*p* < 0.05) and positively (r > 0) with Kf one month after CXL; NGF-β correlates significantly (*p* < 0.05) and positively (r > 0) with CYL nine months after CXL, and, after twelve months, NGF-β correlates significantly (*p* < 0.05) and positively (r > 0) with Ks and Kf (Table 3). We conducted a linear regression analysis to determine if NGF-β, in combination with any other factor, influences the degree of keratoconus. It turned out that all the relationships were statistically insignificant (all the *p*-values were above 0.05).

### 2.3. Corneal Sensitivity

Table 4 represents changes in the corneal sensitivity levels over time, on a scale from 0 to 6, after the procedure. All the results were statistically significant (*p* < 0.001). One day after the procedure, a significant decrease (an average of 5.9) was observed compared to before the procedure, which persisted for approximately one month after the procedure. After three months, the values increased and returned to the baseline levels around 9–12 months after the procedure. Corneal sensitivity did not correlate significantly with the level of NGF-β secretion, and the division into groups based on age and the degree of keratoconus did not have a statistically significant effect on corneal sensitivity changes.

### 2.4. In Vivo Confocal Microscopy (IVCM)

We observed changes occurring in the cornea, particularly in corneal nerves, in the patients with keratoconus both before and after CXL over a period of 12 months. The quantity of corneal nerves correlates with corneal sensitivity, which decreases immediately after surgery but returns to normal after nerve regeneration. In the initial period after the procedure, we observed a significant decrease in corneal nerve density (Figure 3a) due to mechanical damage to the corneal epithelium and the use of UV radiation during the CXL procedure. The first signs of regeneration appeared around 1 month after the procedure (Figure 3b), and the recovery of nerves to their pre-procedure levels varied from 6 months to approximately 1 year (Figure 3c).

## 3. Discussion

The cause of keratoconus and factors predicting the progression of the disease have been a mystery to us to date. Despite numerous reports, there is still no clear answer about the reasons. In recent years, the number of publications on the etiopathogenesis of keratoconus has significantly increased, and the primary topic of discussion is whether it has an inflammatory basis. Analyses of tears and corneal tissue in patients with keratoconus have shown an increase in several cytokines, including IL-6, IL-8, IL-1β, TNF-α, and IFN-β, which may support this hypothesis [6,10,28,29].

In our study, we focused on the analysis of nerve growth factor in patients with keratoconus before CXL and observed changes in the secretion of this factor after the procedure. This is the largest study to date that assesses NGF levels in patients with keratoconus’ tears and the second in terms of observing changes in this parameter after CXL [23].

The nerve growth factor (NGF) was discovered in the 1950s by the Italian neurologist Dr. Rita Levi-Montalcini, who received the Nobel Prize for her achievements in 1986 [30,31]. The NGF was the first neurotrophic factor to be discovered and is the best characterised. The NGF is activated by binding to the tropomyosin receptor kinase A (TrkA) and the low-affinity nerve growth factor receptor (p75NTR) [32]. These receptors are located in the cornea (in cells such as the epithelium, endothelium, and stroma), limbal stem cells, conjunctival epithelial cells, retinal pigment epithelium, photoreceptors, and retinal ganglion cells. The NGF influences the corneal surface’s equilibrium by promoting the proliferation of limbal stem cells, immunomodulation, and tear production, ultimately leading to the proper structure of the corneal epithelium [17]. Disturbance in the NGF secretion pathway can negatively impact the regenerative processes, leading to delayed corneal wound-healing and subsequent opacity.

A. Lambiase studied the molecular basis of keratoconus, focusing on changes in the NGF receptor secretion pathways. This study found a lack of TrkA and a significant decrease in the p75 and NGF levels in the corneas of patients with keratoconus compared to healthy subjects. Interestingly, the corneas with keratoconus parameters induced by refractive surgery and other corneal diseases showed normal levels of TrkA [20]. Using this hypothesis, it can be deduced that individuals with keratoconus undergo a downregulation of NGF secretion, which our research results may confirm. In our study, the NGF level was significantly reduced (8.11) in the patients with keratoconus compared to the control group (16.73), *p* < 0.001. Giusi Precipe also investigated the expression of the NGF receptor TrkA in immune cells of patients with joint inflammation, which showed a reduced expression of TrkA. The author suggests that faulty TrkA expression may facilitate pro-inflammatory mechanisms, contributing to chronic inflammation and tissue damage [33].

On the first day after CXL, the NGF level decreases significantly. We do not know the cause of this decrease, but it may be due to excessive tearing on the day after the procedure, diluting the samples significantly and resulting in a decrease in this parameter. Additionally, damage to the corneal epithelium may cause a drop in NGF secretion immediately after the procedure, and steroid drops further inhibit the rapid secretion of NGF [34]. Subsequently, during the corneal epithelial regeneration process (within the first seven days), a slight increase in NGF was observed, although it was not statistically significant. Long-term changes in NGF levels after CXL were also not detected. Bence Lajos Kolozsvári’s findings were similar [23,35].

An interesting observation is that, when we divided our patients into groups with mild, moderate, and advanced keratoconus, it was found that, in individuals with advanced keratoconus (Ks > 56D), there was a significant increase in NGF secretion in the first month after the procedure compared to the groups with moderate and mild keratoconus. This difference is likely related to greater corneal damage after CXL in patients with advanced keratoconus. Many publications describe that the degree of advancement affects the frequency of complications and worse disease progression inhibition results after CXL [36,37,38].

After other similar minimally invasive procedures in refractive surgery, NGF secretion increased significantly and was dependent on the degree of corneal damage associated with the surgery. The postoperative NGF levels in tears were higher in the FS-LASIK group compared to the smile ReLEx group, and, after PRK, NGF levels were higher than after LASIK [24,39]. This confirms the significant role of NGF in corneal epithelial regeneration and its association with the severity of corneal surface damage.

It is strange that there was no significant increase in NGF secretion after CXL in patients with keratoconus. One possible explanation for this may be a disruption in the reparative response generated by the injured cornea in patients with keratoconus. Cheung IM examined the concentration of interleukin-1α (IL-1α), fibroblast growth factor 2 (FGF-2), nerve growth factor beta (β-NGF), insulin-like growth factor 1 (IGF-1), tumour necrosis factor alpha (TNF-α), epidermal growth factor (EGF), transforming growth factor β1 (TGF-β1), platelet-derived growth factor (PDGF), and hepatocyte growth factor (HGF) in response to secondary damage in keratoconic corneas in a control group. It was found that corneas with keratoconus have a reduced regulation of regeneration, which may make them unable to produce an adequate response to injury. Over time, this can lead to inadequate tissue repair after damage [40]. Under normal conditions, the NGF, in conjunction with receptors on monocytes, reduces the production of inflammatory cytokines (IL-1β, TNF-α, IL-6, and IL-8), while simultaneously inducing the release of anti-inflammatory mediators (IL-10 and IL-1 receptor antagonists). In patients with keratoconus, this regulation is likely disrupted by faulty TrkA receptor expression [33].

Current reports suggest that inflammatory mediators may influence keratoconus progression. Due to the progressive nature of the disease, early diagnosis and appropriate treatment are exceptionally important. In our study, we analysed the correlation of NGF-β with corneal topographic parameters measured using CASIA (K1, K2, CYL, CCT) before and after CXL. It was found that the NGF significantly positively correlates with the K1 parameter before the procedure and one month after. Additionally, it positively correlates with the CYL value after nine months and with K1 and K2 one year after the procedure. These results indicate that the NGF is strongly associated with corneal topographic parameters reflecting the severity of the disease. Similar results were obtained by Bence Lajos Kolozsvári, where the NGF showed a significant positive correlation in all age groups with seven out of the eight investigated PENTACAM topographic indicators (K2, Ave K, KSI, OSI, CSI, KPI, and SDP) [21]. Interesting reports by Fodor M suggest that the level of NGF, in combination with IL-13, can predict corneal keratoconus progression with 100% specificity and 80% sensitivity [22]. This demonstrates the significant role of the NGF in the pathophysiology of keratoconus and emphasises the importance of conducting further research to determine its potential role in disease progression.

We did not find a significant association between corneal sensitivity and the level of NGF secretion in tears, likely due to the disrupted secretion of NGF in the patients with keratoconus. The change in corneal sensation in our study group correlated with the IVCM image and reports of corneal nerve return after CXL treatment. Immediately after the procedure, we observed the absence of nerves, which began to return around 1 month after the procedure, with a return to normal ranging between 6 and12 months [41,42].

Currently, the future of treating conditions with disrupted corneal innervation and dry eye has become the use of recombinant human NGF eye drops. They have been successfully introduced for the treatment of neurotrophic keratopathy and dry eye syndrome. Clinical trials have also explored their application in diseases such as glaucoma and retinitis pigmentosa with cystoid macular oedema. Recently, the safety and effectiveness of rh-NGF have also been demonstrated in children and adolescents [43,44]. The wide range of potential that these drops offer is a great hope for the treatment of various eye diseases.

Gong Q examined the impact of NGF eye drops on corneal sensitivity, regeneration of corneal nerves, and dry eye symptoms in patients after LASIK surgery compared to patients treated with saline (NS) and lubricating eye drops. He noted a significant improvement in these parameters in the patients treated with rh-NGF [45]. However, no one has evaluated the beneficial effects of rh-NGF eye drops in patients with KC after CXL or corneal transplantation. Nevertheless, it can be assumed that they will have a positive impact on epithelial regeneration and the acceleration of corneal sensation recovery, ultimately reducing distant complications such as corneal opacities. This would be particularly important due to the initially lower concentration of endogenous NGF and reduced repair mechanisms in these patients.

## 4. Materials and Methods

### 4.1. Study Population

This study was conducted in accordance with the Helsinki Declaration guidelines and received approval from the Bioethics Committee of the Silesian Medical University in Katowice (PCN/0022/KB1/21/21). Each participant provided informed consent for the examination and study. This study used random sampling among the patients who had given their informed consent to participate in the study from 2021, reaching 52 subjects approved by the bioethics committee.

We included 52 individuals in the prospective randomized study, all of whom were eligible for CXL treatment due to progressing keratoconus. The study consisted of 13 women and 39 men, with an average age of 24.35 (ranging from 11 to 47) years, including 10 participants below the age of 18. Comprehensive ophthalmic examinations were conducted for all the patients, and tear samples were collected before and after CXL during a one-year observation period at regular intervals: 1 day before and after the surgery, at the day 7 visit, and at 1, 3, 6, 9, and 12 months after CXL. The disease progression criteria included changes in corneal topographic parameters, such as an increase in astigmatism or corneal curvature (K1, K2), a maximum corneal curvature (Kmax) increase of >1 D within 12 months from the last follow-up visit, a rise in posterior corneal elevation by >15 µm, and a decrease in visual acuity by one or more Snellen lines in corrected visual acuity. The control group comprised 20 healthy individuals. The exclusion criteria for both the patients with keratoconus and the control group were as follows: the presence of systemic diseases, individuals with dry eye disease (DED), the use of systemic and topical medications, a history of previous eye injuries and/or surgeries, active eye inflammation, allergies in medical history, contact lens wear, pregnancy, and breastfeeding. Those strict criteria for the selection of both the control and study groups were established to create the most homogenous groups possible, aiming to eliminate potential complications in our research results.

### 4.2. Cross-Linking Treatment

The CXL procedure was performed according to the Dresden protocol at the University Hospital Ophthalmology Department of the Railway Hospital in Katowice. All the patients eligible for the procedure had a corneal thickness of >400 µm, making Epi-off (3 mW/cm^2^, 30 min) the preferred treatment. After the procedure, one drop each of levofloxacin (5 mg/mL) and dexamethasone (1 mg/mL) were administered, and a soft bandage contact lens (Air Optix Aqua; Ciba Vision, Alcon, Fort Worth, TX, USA, diameter 14.2, curvature 8.6) was placed, which was typically removed after the corneal epithelium had healed, usually at the first follow-up visit (7 days post-procedure). All the patients used levofloxacin with dexamethasone eye drops and preservative-free artificial tears (based on sodium hyaluronate) five times a day until the first follow-up visit. Subsequently, the antibiotic drops were gradually reduced and used for two weeks, and the steroid eye drops were continued for 1.5 months after the procedure.

### 4.3. Measurements

All the patients underwent a comprehensive ophthalmic examination, including best-corrected visual acuity, slit-lamp examination, intraocular pressure measurement, and corneal topography using anterior segment optical coherence spectral tomography (SS OCT; CASIA2 OCT; Tomey, Nagoya, Japan). Flat (K 1), steep (K 2), cylindrical (CYL), and central corneal thickness (CCT) keratometry were evaluated; corneal sensitivity was measured using a Cochet-Bonnet esthesiometer (Luneau Ophthalmologie, Paris, France), and corneal nerves were assessed using in vivo confocal microscopy (HRT 3, Heidelberg Engineering GmbH, Ger-many). The severity of keratoconus was assessed as follows: mild when the steepest keratometric reading (Ks) was <48 dioptres (D); moderate when Ks ranged from 48 to 54 D; and severe when Ks was >54 D.

### 4.4. Tear Collection and Analysis

Lacrimal fluid was non-invasively collected from both the study and control group participants using a microcapillary from the lower conjunctival sac after gently everting the lower eyelid. Local anaesthesia was not used for this procedure. The collected tears were transferred into Eppendorf-type tubes and immediately stored in a freezer at −80 °C until tear analysis. The collected samples had a volume of approximately 10 µL. The tear analysis using the Bio-Plex technique was performed at the Microbiology and Immunology Department in Zabrze. The thawed tears were centrifuged at 4 °C with an acceleration of 14,000× *g* for 5 min. After centrifugation, the supernatants were transferred to 96F-Well Microplates and diluted with a buffer to a volume of 50 µL. Standards for the measured analytes were also placed in the wells of the plate. Then, a suspension of magnetic beads coated with specific antibodies for the analytes to be measured was added. The subsequent steps were conducted according to the manufacturer’s instructions. Finally, the obtained beads, coated with complexes and marked with phycoerythrin, were analysed using the Bio-Plex 3D Suspension Array System and xPONENT 4.0 aqusition software (Bio-Rad Laboratories Inc., Hercules, CA, USA). The concentrations of the measured cytokines were determined based on standard curves using the manufacturer’s software [46,47].

### 4.5. Statistical Methods

The comparison of the quantitative variables between the study and control groups before the procedure was conducted using the Mann–Whitney test. The Kruskal–Wallis test was used to correlate the NGF and the degree of KC severity. To analyse the relationship between β-NGF levels and keratometric readings and the correlation of corneal sensation with NGF levels, the Spearman correlation coefficient was used. The changes in the parameter values over time were analysed using linear mixed-effects models. The simultaneous influence of two potential predictors on the quantitative variable was analysed using linear regression. The results were presented in the form of regression parameters along with 95% confidence intervals. A significance level of 0.05 was adopted in the analysis. Therefore, all *p*-values below 0.05 were interpreted as indicating significant relationships. The analysis was performed using the R software, version 4.3.1.

## 5. Conclusions

The lack of a clear answer regarding the role of inflammation in keratoconus necessitates further research. This is particularly significant due to the early and insidious onset of the disease, which can lead to substantial visual impairment and affect the quality of life of patients [48]. Tear film analysis is a non-invasive procedure, and, in the future, tear proteomics may prove invaluable in diagnosing keratoconus, monitoring disease progression, and developing targeted therapies. To the best of our knowledge, this is the largest study of its kind aimed at evaluating the NGF in the tear film of individuals with keratoconus. Our results emphasise the strong and, perhaps, even pivotal role of the NGF-β in the pathogenesis of this disease, as a result of its disturbed secretion.

## Figures and Tables

**Figure 1 ijms-25-00366-f001:**
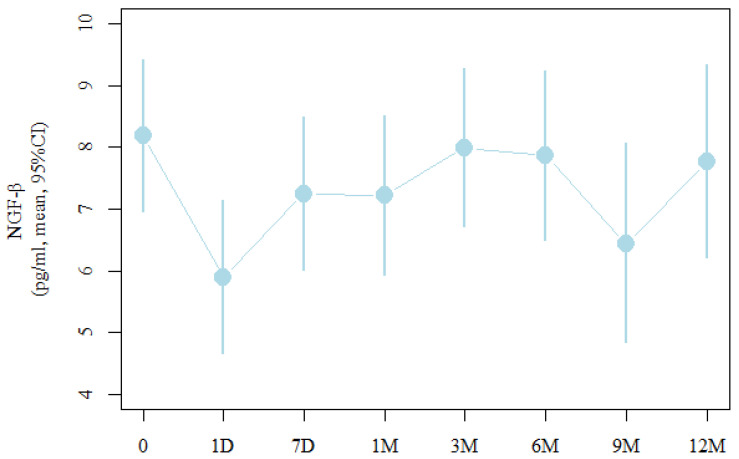
Analysis of the level of NGF-β in patients with KC before CXL treatment (0), one day after CXL (1 D), seven days after CXL (7 D), one month after CXL (1 M), three months after CXL (3 M), six months after CXL (6 M), nine months after CXL (9 M), and twelve months after CXL (12 M).

**Figure 2 ijms-25-00366-f002:**
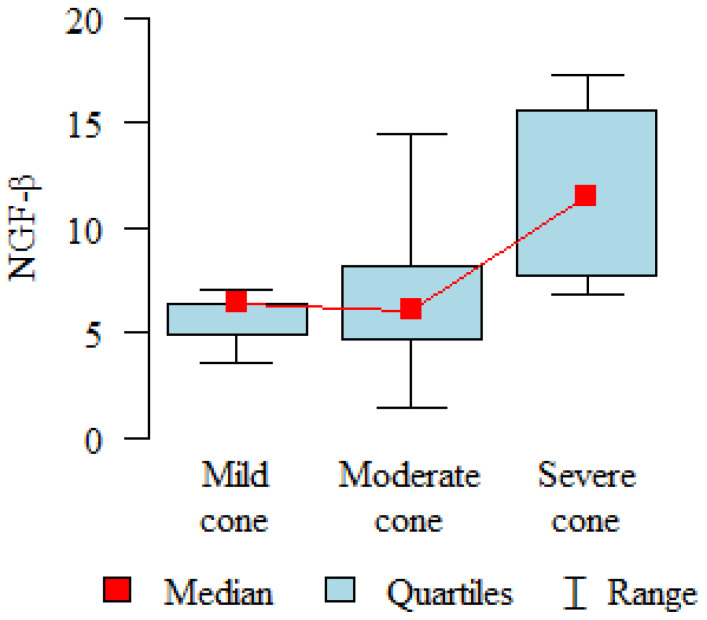
Analysis of NGF-β values one month after CXL divided into the degree of disease advancement (mild, moderate, and severe).

**Figure 3 ijms-25-00366-f003:**
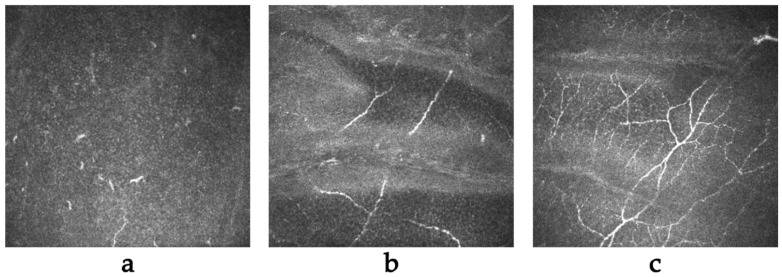
(**a**) Virtually complete loss of nerves of the subepithelial plexus (SEP) immediately after CXL treatment; (**b**) initial regeneration of subepithelial plexus nerves approximately one month after CXL; and (**c**) complete repopulation of the subepithelial plexus nerves approximately six months after CXL treatment.

**Table 1 ijms-25-00366-t001:** β-NGF concentration values in the patients with keratoconus and in the healthy control subjects before CXL procedure and values in the same people when dividing the KC patients by age and stage of disease advancement.

Parameter		Group	N	Moderate	SD	Median	Min	Max	Q1	Q3	*p*
NGF-β	ALL	Studied group	48	8.11	5.86	6.42	3.1	28.87	4.17	9.4	*p* < 0.001 *
		Control group	20	16.73	7.71	14.36	7.21	32.91	10.07	22.28	
	AGE										
	up to 19 years	Studied group	15	8.86	5.9	7.89	3.94	28.87	6.42	9.45	*p* < 0.001 *
		Control group	20	16.73	7.71	14.36	7.21	32.91	10.07	22.28	
	20–26 years	Studied group	16	8.76	6.61	7.39	3.73	28.87	4.09	10.11	*p* = 0.001 *
		Control group	20	16.73	7.71	14.36	7.21	32.91	10.07	22.28	
	over 26 years	Studied group	13	6.03	4.85	4.63	3.1	21.57	3.94	5.53	*p* < 0.001 *
		Control group	20	16.73	7.71	14.36	7.21	32.91	10.07	22.28	
	SEVERITY										
	Mild	Studied group	15	5.94	2.68	4.77	3.51	12.93	3.94	7.15	*p* < 0.001 *
		Control group	20	16.73	7.71	14.36	7.21	32.91	10.07	22.28	
	Medium	Studied group	25	9.3	7.29	7.23	3.1	28.87	4.17	9.93	*p* < 0.001 *
		Control group	20	16.73	7.71	14.36	7.21	32.91	10.07	22.28	
	Severe	Studied group	6	7.61	2.01	7.84	4.18	10.36	7.3	8.2	*p* = 0.004 *
		Control group	20	16.73	7.71	14.36	7.21	32.91	10.07	22.28	

* Statistically significant (*p* < 0.05).

**Table 2 ijms-25-00366-t002:** NGF-β values during the 12-month observation period after CXL in patients with keratoconus.

Measure	Moderate	95%	CI
Before treatment (0)	8.2	7	9.4
Day after the procedure (1 D)	5.9 (*p* = 0.037 *)	4.7	7.1
After seven days (7 D)	7.3	6	8.5
After one month (1 M)	7.2	5.9	8.5
After three months (3 M)	8	6.7	9.3
After six months (6 M)	7.9	6.5	9.2
After nine months (9 M)	6.4	4.8	8.1
After 12 months (12 M)	7.8	6.2	9.3

* Statistically significant (*p* < 0.05).

**Table 3 ijms-25-00366-t003:** Significant correlations between β-NGF release and topographic data in KC.

	Ks	Kf	CYL	PI—Apex	PI—Thinnest
Measurement before treatment	r = 0.265, *p* = 0.075	r = 0.315, *p* = 0.033 *	r = 0.08, *p* = 0.597	r = −0.172, *p* = 0.254	r = −0.222, *p* = 0.138
Measurement after seven days	r = 0.297, *p* = 0.094	r = 0.335, *p* = 0.057	r = −0.01, *p* = 0.956	r = 0.006, *p* = 0.972	r = 0.06, *p* = 0.742
Measurement after one month	r = 0.148, *p* = 0.368	r = 0.32, *p* = 0.047 *	r = −0.092, *p* = 0.579	r = −0.067, *p* = 0.687	r = −0.058, *p* = 0.724
Measurement after three months	r = 0.011, *p* = 0.95	r = 0.021, *p* = 0.899	r = 0.044, *p* = 0.791	r = 0.123, *p* = 0.463	r = 0.025, *p* = 0.883
Measurement after six months	r = 0.059, *p* = 0.765	r = 0.049, *p* = 0.803	r = 0.072, *p* = 0.714	r = −0.138, *p* = 0.483	r = 0.035, *p* = 0.86
Measurement after nine months	r = 0.359, *p* = 0.144	r = 0.066, *p* = 0.794	r = 0.53, *p* = 0.024 *	r = 0, *p* = 1	r = −0.038, *p* = 0.882
Measurement after twelve months	r = 0.493, *p* = 0.012 *	r = 0.525, *p* = 0.007 *	r = 0.143, *p* = 0.495	r = −0.201, *p* = 0.335	r = −0.086, *p* = 0.681

r—Spearman’s correlation coefficient. * Statistically significant dependency (*p* < 0.05).

**Table 4 ijms-25-00366-t004:** Average values of corneal sensation in patients with KC before the procedure and during the 12-month follow-up after CXL.

Group	Moderate	SD
0	6	0
7 D	0.06	0.24
1 M	1.73	0.99
3 M	3.55	1.38
6 M	4.93	1.05
9 M	5	1.22
12 M	5.54	0.85

## Data Availability

The data used to support the findings of this study are included in the article. The data will not be shared due to third-party rights and commercial confidentiality.

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
