# Peer review of "Level of Secretion and the Role of the Nerve Growth Factor in Patients with Keratoconus before and after Collagen Fibre Cross-Linking Surgery"

_ijms, 2023, doi:10.3390/ijms25010366_

Round 1
Reviewer 1 Report
Comments and Suggestions for Authors
The authors have presented a manuscript on "Level of secretion and the role of NGF-β in patients with kera-2 toconus before and after CXL surgery". The manuscript was generally well written, results were well presented and supported by sound statistical analysis.
1.) For the abstract, it will be good if the author can add a short description of the background and problem statement / research gap of the study.
2.) Appreciate if the authors can add objective of the study in the last paragraph of introduction.
3.) Please explain the method used to determine the sample size of 52.
4.) The authors mention that comprehensive ophthalmic examinations were conducted and the disease progression criteria included changes in corneal topographic parameters, such as an increase in astigmatism or corneal curvature (K1, K2), a maximum corneal curvature (Kmax) .... Please briefly describe the procedure and equipment used.
Comments on the Quality of English LanguageMinor editing of English language required
Reviewer 2 Report
Comments and Suggestions for Authors
In the present manuscript (ijms-2776573), entitled "Level of secretion and the role of NGF-β in patients with keratoconus before and after CXL surgery", by Magdalena Krok et al, the authors present their studies referring to the measurement of the level of nerve growth factor (NGF-β) in patients with keratoconus and examined how the level of NGF-β changed after CXL surgery. Studies conducted after similar minimally invasive refractive procedures, including LASIK, confirm that regenerative mediators, including NGF, may play a crucial role in the eye surface repair process after injury. In detail, the authors used tear film analysis, a non-invasive procedure, suggesting that the tear proteomics may prove, in the future, invaluable in diagnosing keratoconus, monitoring disease progression, and developing targeted therapies.
The study may have not been conducted on a large number of the relevant incidences, but, presently, it is the largest. The manuscript is concisely written and well documented.
A minor flaw is that in the Introduction, the information given about the NGF-β is limited. A suggestion would be to include, in the revised version, the following paragraph, so that the NGF-β and its function become more decipherable to the non-cognizant reader.
"Nerve Growth Factor-beta Human Recombinant produced in E. Coli is a non-covalently disulfide-linked homodimer, non-glycosylated, polypeptide chain containing 2 identical 121 amino acids with a molecular weight of two 13.6 kDa polypeptide monomers."
